# Identification of a BTV-Strain-Specific Single Gene That Increases *Culicoides* Vector Infection Rate

**DOI:** 10.3390/v13091781

**Published:** 2021-09-07

**Authors:** Honorata M. Ropiak, Simon King, Marc Guimerà Busquets, Kerry Newbrook, Gillian D. Pullinger, Hannah Brown, John Flannery, Simon Gubbins, Carrie Batten, Paulina Rajko-Nenow, Karin E. Darpel

**Affiliations:** The Pirbright Institute, Ash Road, Pirbright, Woking, Surrey GU24 0NF, UK; honorata.ropiak@reading.ac.uk (H.M.R.); simon.king@pirbright.ac.uk (S.K.); Marc.GuimeraBusquets@pirbright.ac.uk (M.G.B.); kerry.newbrook@pirbright.ac.uk (K.N.); gillian.pullinger@btinternet.com (G.D.P.); hannah_brown90@hotmail.co.uk (H.B.); john.flannery@pirbright.ac.uk (J.F.); simon.gubbins@pirbright.ac.uk (S.G.); carrie.batten@pirbright.ac.uk (C.B.); karin.darpel@pirbright.ac.uk (K.E.D.)

**Keywords:** bluetongue virus, BTV-1, BTV-4, *Culicoides sonorensis*, midge, reassortment, vector infection rate, reverse genetics, KC cells

## Abstract

Since the 2000s, the distribution of bluetongue virus (BTV) has changed, leading to numerous epidemics and economic losses in Europe. Previously, we found a BTV-4 field strain with a higher infection rate of a *Culicoides* vector than a BTV-1 field strain has. We reverse-engineered parental BTV-1 and BTV-4 strains and created BTV-1/BTV-4 reassortants to elucidate the influence of individual BTV segments on BTV replication in both *C*. *sonorensis* midges and in KC cells. Substitution of segment 2 (Seg-2) with Seg-2 from the rBTV-4 significantly increased vector infection rate in reassortant BTV-1_4S2_ (30.4%) in comparison to reverse-engineered rBTV-1 (1.0%). Replacement of Seg-2, Seg-6 and Seg-7 with those from rBTV-1 in reassortant BTV-4_1S2S6S7_ (2.9%) decreased vector infection rate in comparison to rBTV-4 (30.2%). However, triple-reassorted BTV-1_4S2S6S7_ only replicated to comparatively low levels (3.0%), despite containing Seg-2, Seg-6 and Seg-7 from rBTV-4, indicating that vector infection rate is influenced by interactions of multiple segments and/or host-mediated amino acid substitutions within segments. Overall, these results demonstrated that we could utilize reverse-engineered viruses to identify the genetic basis influencing BTV replication within *Culicoides* vectors. However, BTV replication dynamics in KC cells were not suitable for predicting the replication ability of these virus strains in *Culicoides* midges.

## 1. Introduction

Bluetongue (BT) is a non-contagious and infectious disease of domestic and wild ruminants which compromises their health and welfare and is caused by bluetongue virus (BTV). Due to its high socioeconomic impact, it has been labeled as a notifiable disease by the World Organisation for Animal Health (OIE), and suspicion of BT requires reporting to government authorities. Manifestation of BT differs amongst ruminant species with the most severe clinical signs observed in sheep, while cattle are considered a reservoir host, although clinical signs can be observed for some BTV strains [1,2].

BTV is transmitted to its ruminant hosts almost exclusively by *Culicoides* biting midges. Of the known species of *Culicoides* midges (order Diptera, family Ceratopogonidae), only a small number;have been confirmed as biological vectors of BTV, including *C. imicola*, *C. obsoletus* and *C. sonorensis* [3,4,5]. Vector competence is defined as the ability of an arthropod to acquire, become infected with and transmit a pathogen [6,7]. Thus, the competence of *Culicoides* biting midges is a crucial factor in the transmission of BTV between ruminant hosts. Vector competence is commonly expressed as the percentage of individual arthropods that develop a fully transmissible infection following the uptake of a virus-containing blood meal. In the case of *Culicoides* spp., vector competence depends on a range of factors, including the genetic susceptibility of the midge species [8,9,10,11] and external temperature, which affects viral replication/dissemination within the midge [3,12]. BTV genetic factors also play a critical role in *Culicoides* infection efficiency as different BTV strains vary greatly in their ability to infect and replicate, even in the same *Culicoides* species under controlled conditions [13,14,15].

BTV is a non-enveloped virus with a segmented double-stranded RNA (dsRNA) genome, belonging to the *Orbivirus* genus within the *Reoviridae* family. Ten linear genome segments (Seg-1 to Seg-10) encode for seven structural (VP1–VP7) and five non-structural (NS1, NS2, NS3/NS3A, NS4 and NS5) proteins [16,17,18]. The outer capsid of the BTV particle is formed of VP2 and VP5, which allow for membrane attachment and penetration. The VP2 trimers protrude from the outer capsid surface resembling spike-like structures, which facilitate virus attachment to the host receptor. In contrast, the VP5 trimers are less exposed on the virion surface than VP2 [19]. The BTV core is composed of two structural proteins (VP3 and VP7), and it encloses transcription complexes (VP1, VP4 and VP6) in addition to the viral genome. The non-structural proteins play a key role in virus replication, maturation and exit from infected cells [16,17,18].

There are 24 classical BTV serotypes (BTV-1 to BTV-24) which are classified by differences in the variable outer capsid protein, VP2 [20]. VP5 is also involved in serotype determination, to a lesser extent, through influencing the structural conformation of VP2 [21]. Unlike the classical serotypes, which are spread by insect vectors, direct transmission between infected animals has been documented for atypical serotypes such as BTV-26 and BTV-28 [22,23]. The clinical outcome of BT varies according to the virus strain involved: serotype alone does not allow for virulence prediction as virulent and avirulent strains of the same BTV serotype often exist [24,25]. Although a highly virulent BTV-8 strain was responsible for multiple BTV outbreaks in both cattle and sheep in Europe [26,27], several other strains from additional serotypes have displayed lower levels of pathogenicity within the same ecosystem [28,29].

During the last two decades, there has been a dramatic change in the global distribution of BTV, which is particularly evident in Europe, where multiple BTV serotypes and strains have been introduced to previously BTV-free zones [30]. Amongst the BTV serotypes identified in Europe, strains of BTV-1, BTV-4 and BTV-8 have caused the most severe outbreaks and economic losses [31,32,33]. As of the beginning of 2021, animal movement restriction zones have been imposed in several European countries (or some of their regions), where strains of BTV-1, BTV-4 and BTV-8 are circulating separately or in combination with each other [34]. In areas where several different BTV serotypes are simultaneously circulating, co-infection of ruminants with multiple serotypes has been reported previously [35,36,37,38]. The co-circulation of different serotypes or strains presents opportunities for viral reassortment, as BTV strains can exchange genome segments during co-infection of the same ruminant host or *Culicoides* vector. For segmented viruses such as BTV, reassortment may lead to the emergence of virus strains with altered phenotypes, such as changes in ruminant virulence or the infection rate of *Culicoides* vectors [35,39]. Reassortment occurs frequently amongst BTV strains circulating in the field and is an important feature of BTV evolution, with amino acid substitutions more common after reassortment [39]. Our previous work identified a field reassortant BTV-4 MOR2009/07 that had obtained segments from co-circulating BTV-4 and BTV-1 strains. The parental strains showed a vastly diverging ability to replicate in *C*. *sonorensis*, with BTV-4 MOR2004/02 leading to a significant infection rate of *C*. *sonorensis* compared to BTV-1 MOR2007/01. High infection rates of *C*. *sonorensis* were also observed for the BTV-4 MOR2009/07 reassortant, in which Seg-2, -3, -6 and -9 had derived from the BTV-4 strain and all other segments had originated from the co-circulating BTV-1 strain [40].

In order to determine the impact of specific BTV segments on infection efficiency of *Culicoides*, we used a reverse genetics system to generate selected BTV-1/BTV-4 reassortants based on wild-type isolates originating from Morocco, with a specific focus on the serotype determining Seg-2. We also generated reverse-engineered rBTV-1 and rBTV-4 strains representing the parental strains, to establish that the trait of *Culicoides* infection was indeed genetically conferred by these strains and not influenced by other factors such as the cell culture used to generate BTV isolates. These BTV strains were utilized to determine the infection rates in a well-established laboratory colony of adult *C. sonorensis* as well as the replication kinetics in a *Culicoides*-derived cell line (KC).

## 2. Materials and Methods

### 2.1. Cell Lines

BSR cells, clones of BHK-21 cells [41], were propagated in a complete growth medium containing Dulbecco’s Modified Eagle Medium (DMEM, with high glucose, GlutaMAX™ Supplement, pyruvate) supplemented with 5% foetal bovine serum (FBS; heat-inactivated, BTV Ab negative), (Gibco, Paisley, UK) and penicillin-streptomycin (100 U/mL and 100 μg/mL; Sigma-Aldrich, Gillingham, UK) at 37 °C with 5% CO_2_.

KC cells, originally derived from embryonic cells of *C. sonorensis* (formerly *C. variipennis*) [42], were propagated in a complete growth medium containing Schneider’s Insect Medium (with L-glutamine and sodium bicarbonate) supplemented with 10% FBS, penicillin-streptomycin (100 U/mL and 100 μg/mL) and amphotericin B (2.5 µg/mL), (Sigma-Aldrich, Gillingham, UK) at 26 °C.

### 2.2. Viruses

#### 2.2.1. Wild-Type BTV Strains

The wild-type (wt) BTV strains wtBTV-1 MOR2006/06 (cell passage E1/BSR4/BHK2) and wtBTV-4 MOR2004/02 (cell passage E1/BHK4) were obtained from the Orbivirus Reference Collection (ORC) at the Pirbright Institute, UK (https://www.reoviridae.org/dsRNA_virus_proteins/ReoID/BTV-isolates.htm (accessed on 11 April 2020)). Both isolates, labeled as BTV-1 MOR2006/06 and BTV-4 MOR2004/02 in the ORC, originated from two separate BTV outbreaks in Morocco and were isolated from sheep. The full-genome sequencing data of these BTV isolates were previously reported [39] and are available from GenBank (BTV-1 MOR2006/06, Seg-1–Seg-10: KP820889, KP821009, KP821131, KP821251, KP821371, KP821491, KP821613, KP821733, KP821854 and KP821974; BTV-4 MOR2004/02, Seg-1–Seg-10: KP820941, KP821061, KP821183, KP821303, KP821423, KP821543, KP821665, KP821785, KP821905 and KP822026). Virus stocks for in vitro and in vivo experiments were prepared by one additional propagation on BSR cells in a complete growth medium. After harvesting, virus stocks were kept at 4 °C. Virus isolation was confirmed using the BTV real-time RT-qPCR assay targeting Seg-10 [43], with some modifications (see Section 2.3).

#### 2.2.2. Reverse-Engineered and Reassortant BTV Strains

The reverse-engineered BTV-1 and BTV-4 strains rescued in this study were designated as rBTV-1 and rBTV-4 to distinguish them from the wtBTV-1 and wtBTV-4 strains. The nomenclature ascribed to the BTV-1 and BTV-4 reassortant strains is indicated by the BTV backbone and subscripted text referring to the heterologous reassortant segment(s), e.g., BTV-1_4S2_ consists of a BTV-1 backbone with Seg-2 derived from the parental BTV-4 strain.

A previously described reverse genetics system [44], which employs transfection of BSR cells with a mixture of in vitro synthesized T7 transcripts [45], was used to generate BTV reverse-engineered and reassortant strains. Briefly, viral dsRNA, originating from either wtBTV-1 or wtBTV-4 isolates, was extracted from infected BSR cell culture pellets using TRIzol (Invitrogen, Paisley, UK). Generated cDNA of complete-genome segments [46] was amplified by PCR [45]; primers used are listed in Appendix A. The PCR products were then double-digested with restriction enzymes as shown in Appendix A (New England BioLabs, Hitchin, UK). This was followed by ligation into pGEX-4T-2 vector (GE Healthcare, Amersham, UK), digestion with the same restriction enzymes and transformation into MAX Efficiency^®^ Stbl2™ Competent Cells (*E. coli* STBL2; Invitrogen, Paisley, UK). Plasmids from obtained colonies were purified using the QIAprep Spin Miniprep Kit (Qiagen, Manchester, UK) and sequenced using the BigDye Terminator v3.1 Cycle Sequencing Kit (Invitrogen, Paisley, UK) and the ABI 3730 DNA Analyzer system (Applied Biosystems, Paisley, UK). Sequence analysis was carried out using the Lasergene 12 SeqMan Pro (DNASTAR Inc., Madison, WI, USA). Clones of Seg-6 from wtBTV-1 were unstable; therefore, the relevant cDNA copy of this gene containing upstream T7 promoter was synthesized and subcloned by GeneArt Gene Synthesis (Invitrogen, Paisley, UK). Plasmid DNA from each clone was linearized with restriction enzymes (Appendix A) and then transcribed using the mMESSAGE mMACHINE^TM^ T7 Ultra Kit (Invitrogen, Paisley, UK), resulting in capped positive-sense T7 RNA transcripts. To rescue reverse-engineered (*n* = 2) and reassortant (*n* = 7) BTV strains, a two-stage transfection of BSR cells was performed using Lipofectamine 2000 Transfection Reagent (Invitrogen, Paisley, UK) as previously described [45]. The presence of cytopathic effect indicated rescue of BTV strains, which was confirmed by Sanger sequencing. Briefly, TRIzol-extracted dsRNA was converted to cDNA and partially amplified (to 500–1000 bp regions) using SuperScript^®^ III One-Step RT-PCR System with Platinum^®^ *Taq* DNA Polymerase (Invitrogen, Paisley, UK). Sequencing was performed as described above. Rescued reverse-engineered and reassortant strains at passage BSR3 (Appendix A) were deposited in the ORC at the Pirbright Institute, UK (https://www.reoviridae.org/dsRNA_virus_proteins/ReoID/rescued%20BTV.html (accessed on 11 April 2020)). Virus stocks for this study were prepared by one additional propagation on BSR cells in a complete growth medium, resulting in passage BSR4. After harvesting, virus stocks were kept at 4 °C.

### 2.3. RT-qPCR

BTV RNA was extracted from 100 μL of tissue culture supernatant or midge homogenate using the KingFisher™ Flex Purification System and the MagVet™ Universal Nucleic Acid Extraction Kit (ThermoFisher Scientific, Paisley, UK) as described previously [47]. Briefly, 5 μL of sample RNA was analyzed using a BTV-Seg-10-specific RT-qPCR assay [43] adapted to fast cycling conditions [47]. To quantify the level of viral RNA, a log-dilution series of the BTV Seg-10 ssRNA transcript (1 × 10^8^–1 × 10^1^ copies per µL) was included in duplicate on each plate as a standard. The number of BTV genome copies in a sample was determined by comparison of the C_T_ values obtained to the standard curve.

### 2.4. High-Throughput Sequencing

The rBTV-1, rBTV-4 and BTV-1_4S2S6S7_ strains at cell passage BSR4 were processed for sequencing as previously described [48]. Briefly, dsRNA was extracted from cell culture pellets using TRIzol Reagent (Life Technologies, Paisley, UK) and eluted in 100 µL of nuclease-free water (Sigma-Aldrich, Gillingham, UK). Libraries were prepared using the Nextera XT DNA Library Preparation Kit and paired-end-read sequencing was performed using MiSeq Reagent Kit v2 on the MiSeq benchtop sequencer (Illumina, San Diego, CA, USA).

### 2.5. Genome Assembly

Genome assembly was performed as previously described [48]. Briefly, the Trim Galore program 0.6.0 Ed. 2009 (https://github.com/FelixKrueger/TrimGalore (accessed on 10 December 2019)) was used for quality and adapter trimming of FASTQ files (program v0.11.8, http://www.bioinformatics.babraham.ac.uk/projects/fastqc/ (accessed on 12 October 2019)). Then, reads were mapped to a reference using the BWA-MEM tool [49], and the DiversiTools software (http://josephhughes.github.io/DiversiTools/ (accessed on 12 October 2019)) was used to generate the consensus sequence. The full-genome sequences of isolates rBTV-1 and rBTV-4 have been deposited in GenBank under accession numbers MZ065377 through MZ065386 and MZ065387 through MZ065396, respectively.

### 2.6. Virus Titrations

The titer of all viral stocks was determined using KC cells as previously described [50]. Briefly, 10-fold dilutions of each virus stock were prepared in Schneider’s Insect Medium supplemented with penicillin-streptomycin (100 U/mL and 100 μg/mL) and amphotericin B (2.5 µg/mL). Each dilution was then added in quadruplicate into a 96-well plate containing KC cells (in a complete growth medium) at a 1:1 ratio. After 5 days incubation at 27 ± 1 °C, the cells were fixed and labeled [50]. Immunofluorescence of infected cells was visualized on an Olympus CKX53 microscope (Olympus, Southend-on-Sea, UK). The tissue culture infective dose at 50% (TCID_50_) was calculated using the Spearman–Kärber equation [51]. The multiplicity of infection (MOI) was calculated using the estimate associated with plaque-forming units (PFU) [44]: 1 TCID_50_ = 0.7 PFU/mL, which was applied to MOI = PFU/number of cells [52]. Each virus titer was determined as the mean of 3–4 independent titrations. The titrations were repeated at 6-month intervals and prior to oral infection of *C*. *sonorensis*.

### 2.7. Oral Infection of C. sonorensis with BTV

Oral BTV infection of adult *C*. *sonorensis* was performed in vivo as previously described [53], with some modifications. Briefly, newly emerged *C*. *sonorensis* obtained from a laboratory colony [54] maintained at the Pirbright Institute, UK, were fed on defibrinated horse blood (TCS Biosciences, Botolph Claydon, UK) containing a strain of BTV at a ratio of 2:1 (blood:virus) using the Hemotek system (Hemotek Ltd., Blackburn, UK). Virus strains tested were rBTV-1, rBTV-4, BTV-1_4S2_, BTV-1_4S2S6S7_ and BTV-4_1S2S6S7_. Each virus was diluted to 5.9 log_10_ TCID_50_/mL in Schneider’s Insect Medium before being mixed with blood. After feeding for 30 min, 16 engorged female midges for each BTV strain were collected, individually homogenized and analyzed using RT-qPCR to determine a baseline C_T_ value for uptake of BTV RNA (day 0). The remaining engorged females were incubated at 25 °C under 70–90% humidity for 8 days. Midges surviving this extrinsic incubation period were individually homogenized by placing each into a separate tube containing a 3 mm stainless steel ball and 200 µL of Roswell Park Memorial Institute 1640 Medium (RPMI; Gibco, Paisley, UK) supplemented with amphotericin B (5 µg/mL) and penicillin-streptomycin (200 U/mL and 200 μg/mL). Homogenization (25 Hz, 2 × 30 s, room temp.) was performed with a Tissuelyser™ (Qiagen, Manchester, UK) [53]. Each homogenate was made up to a final volume of 1 mL with RPMI Medium for RNA extraction.

For clarity we will be using the term “vector infection rate” to describe the replication efficiency of the BTV strains assessed in this study.

Vector infection rate (expressed as % of individuals successfully infected with BTV following the uptake of viraemic blood) was assessed by processing, on average, 107 blood-fed midges (range 96–139) for each BTV strain. An individual *Culicoides* was considered “successfully infected” if the quantity of BTV genome detected post-incubation was greater than the average quantity measured immediately after the viraemic blood meal (day 0).

### 2.8. Replication Kinetics

The assessment of virus replication kinetics was carried out in vitro for wtBTV-1, wtBTV-4, rBTV-1 and rBTV-4; selected reassortants: BTV-1_4S2_, BTV-1_4S9_, BTV-4_1S9_, BTV-1_4S2S6S7_ and BTV-4_1S2S6S7_. These reassortants were selected based on the number of radical amino acid substitutions (>4) identified between corresponding segments in rBTV-1 and rBTV-4. In addition, two BTV-1_4S3_ and BTV-4_1S3_ strains were included as a control as the Seg-3 of BTV-1 and Seg-3 of BTV-4 were almost identical. Replication of BTV strains in KC cells was monitored using Seg-10 RT-qPCR over a 10-day period. Initially, KC cells were seeded in 24-well plates at 0.75 × 10^6^ cells/well in a complete growth medium [52] 18 h prior to infection to allow attachment of cells. The medium was removed, and the cell monolayer was infected at MOI = 0.05 (in 0.3 mL of medium/well). Three replicates (wells) were used per time point, including 3 negative control wells per plate. Inoculated plates were incubated at 4 °C for 1 h to allow adsorption of virus and synchronization of virus entry [55]. After incubation, wells were washed 3 times with phosphate-buffered saline (PBS, free of Mg^2+^ and Ca^2+^), and 1 mL of complete growth medium was added to each well afterwards. The cell culture supernatant was collected from 3 wells and designated as 0 days post-infection (dpi). The plate was incubated further at 27 ± 1 °C and supernatant was collected at 1, 2, 3, 6, 7 and 10 dpi. Evaporation was minimized by adding PBS to empty wells and by using a plate sealer. The negative control was collected at 10 dpi. The cell culture supernatant from each time point was stored at 4 °C until the end of the experiment and then the BTV was quantified using RT-qPCR as described earlier. Replication kinetics were determined in at least one independent experiment with three replicates for each dpi (see Appendix A).

### 2.9. Statistical Analysis

#### 2.9.1. Vector Infection Rate

Vector infection rate was compared amongst BTV strains using a binomial-family generalized linear model with a logit link function. The number of positive insects (defined as those tested at day 8 having a genome copy value greater than the median of those tested at day 0) and the number of insects tested were the response variables and BTV strain was the explanatory variable. The analysis was implemented in R (version 4.0.5) [56].

#### 2.9.2. Replication Kinetics in KC Cells

Data on viral replication of different BTV strains in KC cells were modeled using logistic replication curves. More specifically, the log_10_ copy number (*Y_s_*(*t*)) for strain *s* at time *t* is given by
(1)$$Y_{s}(t)=\kappa_{1}^{\left(s\right)}+\frac{\left(\kappa_{2}^{\left(s\right)}-\kappa_{1}^{\left(s\right)}\right)}{1+J_{s}\exp\left(-\beta_{s}t)\right)},$$
where *β_s_* is the replication rate,
$$J_{s}=\frac{\kappa_{2}^{\left(s\right)}-\kappa_{1}^{\left(s\right)}}{Y_{0}^{\left(s\right)}-\kappa_{1}^{\left(s\right)}}-1,$$
and *Y*_0_, *κ*_1_ and *κ*_2_ are the initial value, lower asymptote and upper asymptote for the log_10_ copy number, respectively. The time of maximum replication is given by *δ_s_* = log(*J_s_*)/*β_s_*.

Differences amongst BTV strains were incorporated by assuming a hierarchical structure for the model parameters, so that the parameters for BTV strains are drawn from higher-order distributions, namely,
$$\begin{array}{c} \beta_{s}\sim \mathrm{Gamma}(a_{\beta },b_{\beta }),\\ \kappa_{i}^{\left(s\right)}\sim N(\mu_{\kappa_{i}},\sigma_{\kappa_{i}}^{2}),\\ Y_{0}^{\left(s\right)}\sim N(\mu_{Y_{0}},\sigma_{Y_{0}}^{2}), \end{array}$$
where the *μ*s, *σ*s, *a*s and *b*s are higher-order parameters. If and how the parameters varied amongst BTV strains was explored by considering the fit of the models when no, one, two, three or four of the parameters varied amongst strains. In total, sixteen models were considered (Appendix A).

Parameters were estimated in a Bayesian framework. A normal likelihood was used for the data with expected values given by Equation (1) and error variance, *σ_e_*^2^. Priors for the strain-specific parameters were given by the higher-order distributions. Normal priors (with mean 0 and standard deviation 10) were used for the *μ*s, exponential priors (with mean 100) were used for the *σ*s and exponential priors (with mean 1) were used for *a_β_* and *b_β_*. The methods were implemented in OpenBUGS (version 3.2.3; http://www.openbugs.info (accessed on 11 April 2020)). Two chains each of 100,000 iterations were generated (with the preceding 50,000 iterations discarded to allow for burn-in of the chains). Chains were subsequently thinned (by selecting every tenth iteration) to reduce autocorrelation amongst the samples. Convergence of the chains was monitored visually using the Gelman–Rubin statistic in OpenBUGS. Different models for the variation amongst strains in parameters were compared using the deviance information criterion (DIC) [57].

To assess whether the replacement of individual segments (e.g., rBTV-1 vs. BTV-1_4S9_) had an impact on the replication kinetics in KC cells, differences between BTV strains in the replication rate and time of maximum replication were assessed by comparing the posterior distributions for each parameter. Specifically, the proportion of posterior samples for which the parameter for one strain was less than that for the other strain was calculated. If the proportion was <0.05, the parameter for the first strain was considered to be significantly greater than the second. Conversely, if the proportion was >0.95, the parameter for the first strain was considered to be significantly lower than the second (Appendix A).

## 3. Results

### 3.1. Generation of Plasmid Clones

The majority of genome segments derived from wtBTV-1 and wtBTV-4 were cloned into the pGEX-4T-2 vector. Cloning was confirmed by Sanger sequencing and the inserts with the lowest number of nucleotide substitutions in comparison to the wtBTV strains were selected for the generation of reverse-engineered and reassortant BTV strains. However, clones of Seg-6 from wtBTV were unstable; therefore, a plasmid clone was synthesized and subcloned instead.

### 3.2. Generation of Reverse-Engineered and Reassortant BTV Strains

A well-established reverse genetics system [44] was used to rescue reverse-engineered/reassortant BTV strains. Both reverse-engineered viruses rBTV-1 (derived from wtBTV-1 MOR2006/06) and rBTV-4 (derived from wtBTV-4 MOR2004/02) were rescued, which was confirmed by full-genome sequencing. Mono-reassortants (*n* = 5) were generated between rBTV-1 and rBTV-4 reverse-engineered viruses as confirmed by partial Sanger sequencing (Appendix A). Two of the mono-reassortants (BTV-1_4S6_ and BTV-4_1S2_) could not be rescued despite numerous attempts. Instead, two triple reassortants were generated, designated as BTV-4_1S2S6S7_ and BTV-1_4S2S6S7_, containing the heterologous Seg-2, Seg-6 and Seg-7 in both reverse-engineered viruses (Appendix A).

### 3.3. Full-Genome Sequencing

To compare the absence of critical genome changes between wt strains (wtBTV-1 and wtBTV-4) and their reverse-engineered counterparts (rBTV-1 and rBTV-4) the complete genomes were determined by high-throughput sequencing. The coding regions of reverse-engineered strains rBTV-1 (passage BSR4) and rBTV-4 (passage BSR4) were compared to those of wtBTV-1 (passage E1/BSR4/BHK2) and wtBTV-4 (passage E1/BHK4) strains, respectively. For BTV-1, wt and reverse-engineered strains were almost identical; only one conservative substitution at Seg-6 (VP5, L450V) was recorded. For BTV-4, wt and reverse-engineered strains differed by one radical amino acid (aa) substitution related to two segments, Seg-6 (VP5, L521R) and Seg-7 (VP7, E266V). Both reverse-engineered strains (rBTV-1 and rBTV-4) had a low aa% similarity in the VP2 (62.37%) and VP5 (79.32%) coding regions. A high aa% similarity, ranging from 96.10% to 99.42%, was identified in the coding regions of the remaining segments. The aa substitutions in these regions were investigated to determine whether they were a radical or conservative change (Table 1). For two segments, Seg-3 and Seg-5, only conservative aa substitutions were identified, suggesting that these had a minor impact on the vector infection rate observed.

Full-genome sequencing of BTV-1_4S2S6S7_ (passage BSR4) revealed that this reassortant shared 100% nucleotide sequence identity with rBTV-1 (Seg-1, -3, -4, -5, -8, -9 and -10) and rBTV-4 (Seg-2), whereas two radical substitutions were identified in Seg-6 (VP5, R521L) and Seg-7 (VP7, V266E) in comparison to rBTV-4.

### 3.4. Oral Infection of C. sonorensis

To determine which BTV segments affect vector infection rate, a subset of generated reverse-engineered/reassortant BTV strains were used. The estimated vector infection rate for each of the BTV strains tested is shown in Figure 1. The strains were divided into two groups: high vector infection rate (rBTV-4 and BTV-1_4S2_) and low vector infection rate (rBTV-1, BTV-4_1S2S6S7_ and BTV-1_4S2S6S7_). The vector infection rates within each group did not differ significantly (*p* > 0.87) from one another, but they did differ significantly (*p* < 0.002) between the groups.

Two of the three reassortants showed a marked difference in vector infection rate to their corresponding reverse-engineered strain. The introduction of segments Seg-2, Seg-6 and Seg-7 from BTV-1 into BTV-4 resulted in a ~10-fold decrease in vector infection rate from 30.2% to 2.9%. In contrast, the introduction of Seg-2 alone from BTV-4 into BTV-1 resulted in a ~30-fold increase in vector infection rate. In contrast to BTV-1_4S2_, the introduction of Seg-2 along with Seg-6 and Seg-7 from BTV-4 into BTV-1_4S2S6S7_ did not result in a higher vector infection rate.

### 3.5. Replication Kinetics

Fitting different replication curves to BTV replication kinetic data in KC cells showed that the lower asymptote, the initial number of copies and the replication rate all differed significantly amongst BTV strains wtBTV-1, wtBTV-4, rBTV-1, rBTV-4, BTV-1_4S2_, BTV-1_4S3_, BTV-4_1S3_, BTV-1_4S9_, BTV-4_1S9_, BTV-1_4S2S6S7_ and BTV-4_1S2S6S7_ (Appendix A). However, the upper asymptote (i.e., the maximum number of copies produced) did not differ significantly amongst strains (Appendix A). The fitted replication curves for each strain are shown in Figure 2 and the estimated replication rate (*β_s_*) and time of maximum replication (*δ_s_*) for each strain are shown in Figure 3. Further pairwise comparison pinpointed statistically significant differences in replication rate (*p* < 0.05) and time of maximum replication (*p* < 0.05) among BTV strains (Appendix A).

Although we observed differences in genome copies in the initial stages of replication among tested BTV strains, the log_10_ genome copy numbers for all strains reached almost the same level after 6 dpi (~10^6^–10^7^ genome copies/mL), as shown in Figure 2 and Appendix A.

An initial assessment was performed to determine if reverse-engineered BTV strains replicate comparably to their corresponding wt strains (i.e., rBTV-1 vs. wtBTV-1 and rBTV-4 vs. wtBTV-4). There was a significant difference between wtBTV-1 and rBTV-1 in both replication rate (*p* = 0.003) and the time of maximum replication (*p* < 0.001), even though their full-genome sequence was almost identical (i.e., only one conservative substitution in Seg-6). rBTV-1 exhibited a higher replication rate as well as a later time of maximum replication rate. For BTV-4, there was no significant difference in the replication rate or time of maximum replication between wtBTV-4 and rBTV-4, despite two radical aa substitutions related to Seg-6 and Seg-7. There was a significant difference in both parameters between wtBTV strains, with a higher replication rate and a later time of maximum replication for wtBTV-4 (~3 dpi for wtBTV-1 vs. ~5 dpi wtBTV-4). As for reverse-engineered strains, rBTV-4 had significantly lower replication rate in comparison to rBTV-1, and also had a later time of maximum replication.

There were some differences visible in the replication rate and the time of maximum replication of rBTV-1 compared to BTV-1_4S2_ (Figure 3); however, they were not significant (*p* = 0.05). This finding contrasts with our in vivo midge experiment, where a significant increase (30-fold) in vector infection rate was observed after replacement of Seg-2 from BTV-4 into a BTV-1 backbone (i.e., BTV-1_4S2_). Furthermore, the replication rate differed significantly between BTV-1_4S2S6S7_ and rBTV-1, though the time of maximum replication did not. This is also in contrast with our in vivo study, where both strains were placed within the same low vector infection rate group. There was no significant change in the replication rate of rBTV-4 compared to all reassortant strains (except BTV-4_1S2S6S7_, which was significantly lower), and the time of maximum replication of rBTV-4 was significantly later (~5 dpi) than that of all reassortant strains tested.

Moreover, both selected control strains rBTV-1_4S3_ and rBTV-4_1S3_ had significantly earlier times of maximum replication (~3 dpi) compared to their corresponding reverse-engineered strains, rBTV-1 and rBTV-4 (~5 dpi). However, VP3 had high aa% similarity and there was only one conservative substitution present in Seg-3 for both reverse-engineered strains.

In summary, the KC cell line was not a suitable indicator of vector infection rate, as we were not able to distinguish between replication kinetics of selected wild-type, reverse-engineered or reassortant BTV strains under the experimental conditions used in this study. The results from in vitro work did not clearly align to sequencing and in vivo studies for key reassortants BTV-1_4S2_, BTV-1_4S2S6S7_ and BTV-4_1S2S6S7_.

## 4. Discussion

Numerous factors play an important role in determining the ability of BTV strains to infect and subsequently be transmitted by their *Culicoides* vectors. Even the same *Culicoides* species can demonstrate a highly divergent vector competence to different BTV strains [11,14,15,24,58]. However, the genetic basis or influence of specific BTV genes which determine that certain strains replicate better than others in *Culicoides* spp. needs to be further elucidated. Generating reassortant viruses to exchange selected genome segments by reverse engineering represents a powerful tool with which to explore gene traits in segmented viruses. Nonetheless, identifying the genetic basis of phenotypic traits such as the virulence and vector competence observed in wild-type strains has remained challenging for BTV. Such characteristics have proven to be determined by multiple segments and indeed differ across viral strains and (model) host systems [52,59,60,61,62,63].

In this study, we demonstrated that two reverse-engineered BTV strains, based on BTV-1 and BTV-4 derived from the Moroccan outbreaks, retained their diverging infection rates of *C. sonorensis* first observed in wild-type isolates. Full-genome sequences of both reverse-engineered strains, rBTV-1 and rBTV-4, were almost identical to corresponding wild-type BTV stains and all strains replicated effectively in KC cells. Our previous work had identified that the wild-type BTV-4 strain replicated much more efficiently in *C. sonorensis* than a co-circulating wild-type BTV-1 strain. Most interestingly, a field reassortant (BTV-4 MOR2007/09) isolated from the same Moroccan ecosystem, which had obtained four segments from the wild-type BTV-4 (Seg-2, -3, -6 and -9) and all remaining segments from the wild-type BTV-1, showed a much higher vector infection rate of *C. sonorensis* than the wild-type BTV-1 strain did [40].

We generated selected reassortant viruses comprising the two Moroccan BTV-1 and BTV-4 strains to identify which of the genes from BTV-4 MOR2004/02 conferred the replication advantage observed in *C. sonorensis*. We demonstrated that a single gene VP2, derived from the BTV-4 strain, conferred a significant replication advantage in *C. sonorensis*. Equally high vector infection rates were achieved for both rBTV-4 (30.2%) and reassortant BTV-1_4S2_ (30.4%), indicating that a single Seg-2 exchange is sufficient to significantly alter vector infection rate. The replacement of Seg-2 in BTV-4 together with Seg-6 and Seg-7 originating from BTV-1 dramatically reduced vector infection rate in the BTV-4_1S2S6S7_ reassortant (2.9%), confirming the major role of the BTV-4 VP2 in the increased vector infection rates of these two strains. It would be interesting to check in future studies if other strains of BTV-4, circulating in Europe since 2000s, are also characterized by high infection rates of *C. sonorensis*, or if this is a genetic trait encoded by the VP2 of the BTV-4 MOR2004/02 strain only. Infection of adult *C. sonorensis* with different BTV-4 strains, belonging to western (e.g., Morocco, Spain, Portugal and Algeria; 2003–2005) and eastern topotypes (e.g., Greece, Hungary, Slovenia, Croatia, Romania, Italy, France and Corsica; 2014–2021), would help to elucidate if the high infection rates of *C. sonorensis* are serotype, topotype or strain specific. Furthermore, it would also be desirable to assess the replication ability of such BTV-4 strains in other vector species of *Culicoides*, especially as a recent study reported low to moderate vector competence of geographically different populations of *C. obsoletus* and high vector competence of *C. imicola* for the field reassortment isolate BTV-4 MOR2007/09 [14].

Surprisingly, the BTV-1_4S2S6S7_ reassortant was characterized by low vector infection rate (3.0%) despite the presence of Seg-2, Seg-6 and Seg-7 originating from rBTV-4. Therefore, full-genome sequencing was performed and revealed the presence of two radical substitutions in Seg-6 (VP5, R521L) and Seg-7 (VP7, V266E), but no changes were present in Seg-2 in comparison to rBTV-4. It has been shown previously that even a single aa substitution in VP2 (in KC-adapted rescued BTV-11(S1^26^)) can affect vector competence [11]. The substitutions of aa in Seg-6 and Seg-7 of BTV-1_4S2S6S7_ were the same as in wtBTV-4 (cell passage E1/BHK4), possibly indicating a BTV adaptation to a mammalian cell line. It was previously reported that multiple repassage of BTV strains in a mammalian cell culture can alter virulence [61] or the expression of the NS3/NS3a protein [64]. However, it is also possible that the mechanism behind vector competence involves interactions between the capsid proteins VP2, VP6 and VP7 in this BTV-1_4S2S6S7_ reassortant; these molecular interactions could influence the conformation of the receptor binding site.

Within this study, two mono-reassortants that might have further elucidated the genetic basis for the high vector competence of the BTV-4 strain (i.e., BTV-1_4S6_ and BTV-4_1S2_), could not be rescued from cell culture despite numerous attempts; therefore, two triple reassortants were generated instead (designated as BTV-4_1S2S6S7_ and BTV-1_4S2S6S7_). This suggests that some combinations of outer capsid proteins (VP2 and VP5) from different strains might be incompatible [65]. Similar results were reported earlier when a recovery of a mono-reassortant containing Seg-2 of BTV-26 was unsuccessful, but a triple reassortant was generated instead [44].

Comparing the replication kinetics of different BTV isolates in *Culicoides*-derived KC cells with in vivo midge vector infection rates was inconclusive. All BTV strains used in this study efficiently infected and replicated in KC cells. Although the initial number of copies and the replication rate all differed significantly amongst them, there was no apparent pattern. KC cells have been used to identify the genome segments (i.e., Seg-1, Seg-2, Seg-3 and Seg-7) that limit BTV-26 replication in vitro [44]. In another study, an agreement was found between a replication of BTV-11(S10^del^) in KC cells and in vivo experiment. A recombinant BTV-11(S10^del^), which had a small in-frame deletion of 72 aa in NS3/NS3a, replicated to a limited extent in KC cells and failed to propagate/transmit by a colony of *C. sonorensis* [11]. It has been established that propagation of BTV in cell lines can change BTV phenotypes [66], and that subsequent virus passages in cell cultures can lead to adaptive changes and the alteration of replication kinetics in KC cells [64]. Therefore, prior to infections of *C*. *sonorensis* or KC cells in this study, we selected the lowest BSR cell passage possible without any previous history of propagation in KC cells to avoid the introduction of adaptive changes. It is unlikely that differences in replication kinetics were not identified in the KC study due to experimental conditions, as similar experimental design was successfully used to identify key BTV segments for the BTV-26 replication [52]. This outcome deems KC cells as an unsuitable model for determination of vector infection rate for the selected BTV strains, especially ones that can replicate efficiently within KC cells.

## 5. Conclusions

Our results demonstrated that we could utilize reverse-engineered BTV to identify the genetic basis for wild-type BTV viral characteristics. We demonstrated that a single genome segment, namely Seg-2 of the respective Moroccan BTV-4 strain, conferred a significant replication advantage in the *Culicoides* vector.

All BTV strains used in this study efficiently infected and replicated in KC cells to varying degrees without any discernible pattern. We found that BTV replication dynamics in KC cells are not a suitable predictor of BTV replication in adult *Culicoides*.

## Figures and Tables

**Figure 1 viruses-13-01781-f001:**
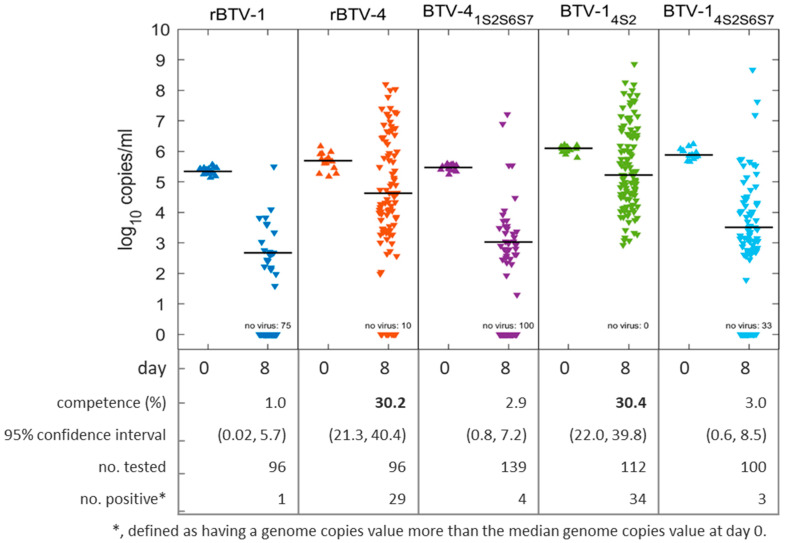
Concentration of BTV genome copies detected in *C. sonorensis* after feeding on blood spiked with reverse-engineered and reassortant BTV strains. Individuals were tested either immediately after feeding (day 0) or after incubation for 8 days at 25 °C (day 8). Symbols depict the concentration of BTV expressed in log_10_ genome copies/mL observed for individuals, and horizontal bars depict the median log_10_ genome copies/mL (excluding those individuals for which no BTV RNA was detected). Comparison between strains was carried out using a binomial-family generalized linear model with a logit link function. *Note*: see Appendix A, if C_T_ values are required for comparison.

**Figure 2 viruses-13-01781-f002:**
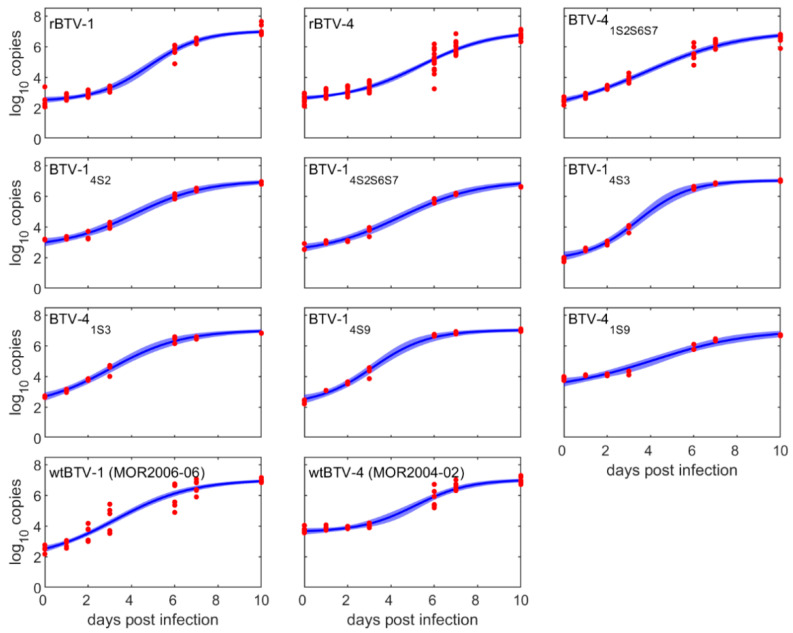
Observed and expected changes in genome copies for eleven BTV strains over time in KC cells. Each plot shows the observed log_10_ genome copy numbers (red circles), posterior median (solid blue line) and 95% credible interval (blue shading) for the fitted replication curves, Equation (1).

**Figure 3 viruses-13-01781-f003:**
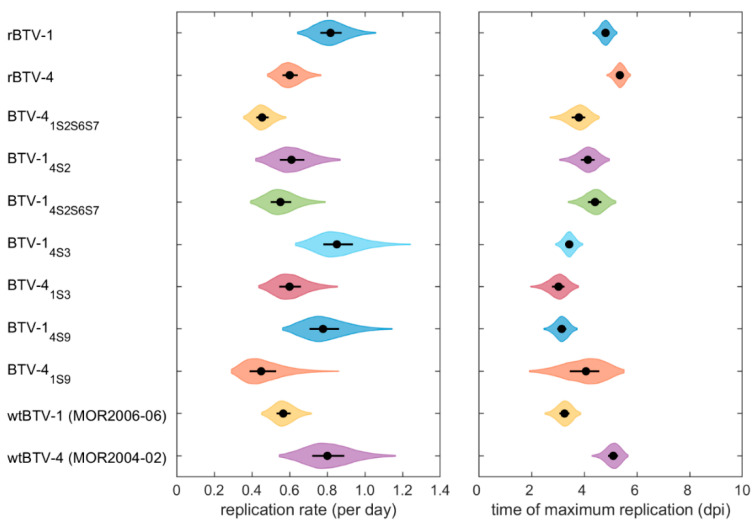
Estimated replication rates and times of maximum replication for eleven BTV strains in KC cells. Violin plots show the posterior median (black circle), interquartile range (black line) and marginal posterior density (shape); dpi, days post-infection.

**Table 1 viruses-13-01781-t001:** Amino acid (aa) differences between reverse-engineered rBTV-1 and rBTV-4 strains for high aa similarity segments.

Genome Segment (Encoded Protein)	Position of aa	Difference in aa		Type of aa Substitution
		rBTV-1	rBTV-4	
Seg-1 (VP1)	55	Q	R	Radical
	179	D	N	Conservative
	1217	T	S	Conservative
	1254	M	I	Radical
Seg-3 (VP3)	241	R	K	Conservative
Seg-4 (VP4)	75	N	S	Radical
	165	E	K	Radical
	239	N	D	Conservative
	305	I	V	Conservative
	343	R	K	Conservative
	355	D	G	Radical
	368	K	R	Conservative
Seg-5 (NS1)	66	R	K	Conservative
	149	I	V	Conservative
	491	K	R	Conservative
Seg-7 (VP7)	266	E	V	Radical
	328	A	V	Conservative
Seg-8 (NS2)	29	Q	L	Radical
	235	G	D	Radical
Seg-9 (VP6)	5	M	I	Radical
	14	K	M	Radical
	30	V	A	Conservative
	37	N	D	Conservative
	51	A	V	Conservative
	95	G	R	Radical
	112	G	R	Radical
	215	H	Q	Radical
	232	S	P	Radical
Seg-10 (NS3)	186	K	R	Conservative
	214	T	A	Radical

## Data Availability

The full-genome sequences presented in this study are openly available in GenBank (see Section 2.5).

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
