# Peer review of "Identification of a BTV-Strain-Specific Single Gene That Increases Culicoides Vector Infection Rate"

_viruses, 2021, doi:10.3390/v13091781_

Round 1

Reviewer 1 Report

I would like to thank the editor and the authors for the opportunity to review such a well-written manuscript. Here, Ropiak and colleagues use reverse engineering to assess how segment reassortment in BTV contributes to viral replication both in cell lines and in vivo. I enjoy how much attention was paid to details in this manuscript. The authors make sure to include all relevant information in the supplementary material while making the text accessible to a broad audience. Language aside (detailed provided below), the experimental design and results, for the most part, support the conclusions of this manuscript. That said, I have a few comments and concerns that should be addressed prior to acceptance for publication.

Major concern:

I respectfully disagree with the authors of this study in using the term vector competence to define their work in both the title and throughout almost the entire manuscript. Vector competence implies the ability of a vector to be infected and transmit a certain pathogen, as referenced by the authors in their intro. That said, checking for viral replication by analyzing the whole body of individual midges is not an accurate measure to imply vector competence. The existence of factors such as midgut and salivary glands infection and scape barriers, host immunity,  etc., are components that ultimately interfere with the ability of a virus to be transmitted by the host and that are ignored when choosing to perform whole-body analyses rather than dissection and quantification of viral particles in a tissue-specific manner, with the inclusion of measuring viral particles in saliva expectorates as gold standard procedure in vector competence assays. Authors are strongly encouraged to remove this term from their title and throughout the manuscript, adapting accordingly. The conclusion provides a good example of how the language should be, rather than oversimplifying the concept of vector competence.

Line (L): 106-107: could the authors please include a few referenced sentences on how common the issue of co-infection in ruminants is?

L 127-129/L172-177: These are good examples of how much attention the authors paid to their text, to make it accessible.

L.155: could the authors please confirm the link? I got an error message saying the URL cannot be found, although the reoviridae.org domain still works.

L.  284: Could the authors please elaborate on what is the commonly reported EIP for wtBTV under this temperature condition, to understand if 8dpi is a good collection point or not.

L 316: “One well as used…” typo?

L436 – Table 1: Could the information for VP2 be also included in this table?

L454 – Figure 1: Can the authors please elaborate on why wtBTV1 and 4 were not included in parallel, as a reference for comparing viral replication efficiency between rescued and reassortant strains? I assume the authors decided to use this approach based on their %aa similarity results between rBTVs and wtBTVs, but given the importance of VP2 is to viral entry and attachment, and its low aa% similarity reported, wouldn't it be ideal to have had this included in the in vivo study as it was for the cell kinetics assay? This comment is also based on the results reported on L492-507 in which differences were observed for BTV-1. On a minor note, I would suggest moving away from the yellow color, as it makes it really hard to visualize the individual data points. I use and recommend a website called www.coolors.co as a really nice alternative to select for visually appealing color combinations/data viz.

L.476 – Figure 2:

Pt.1: Would be interesting if the authors had included 8dpi in their cell kinetics study to match the timepoint in which midges were collected to assess viral load. Any reasons why this was not performed?

Pt. 2: Can the authors please re-arrange this to first display the strains used to infect the midges, next to each other, and separated from the remainder, as to facilitate visual comparisons?

L. 481 – Figure 3: I understand that the results show a maximum replication rate <10dpi, but for consistency, could the X-axis be up to 10 dpi to reflect all collection points?

Table S4: Could the authors provide the results using the scientific notation (x power of 10) to facilitate the visualization?

Author Response

Response to Comments of Reviewers

Manuscript ID: viruses-1356271
Type of manuscript: Article
Title:
Identification of a BTV strain specific single gene that increases Culicoides vector infection rate

We would also like thank to the reviewers for their valuable comments, which have been largely implemented and improved this manuscript. Please find our replies to the specific comments (in italics).

Reviewer 1:

I would like to thank the editor and the authors for the opportunity to review such a well-written manuscript. Here, Ropiak and colleagues use reverse engineering to assess how segment reassortment in BTV contributes to viral replication both in cell lines and in vivo. I enjoy how much attention was paid to details in this manuscript. The authors make sure to include all relevant information in the supplementary material while making the text accessible to a broad audience. Language aside (detailed provided below), the experimental design and results, for the most part, support the conclusions of this manuscript. That said, I have a few comments and concerns that should be addressed prior to acceptance for publication.

Major concern:

I respectfully disagree with the authors of this study in using the term vector competence to define their work in both the title and throughout almost the entire manuscript. Vector competence implies the ability of a vector to be infected and transmit a certain pathogen, as referenced by the authors in their intro. That said, checking for viral replication by analyzing the whole body of individual midges is not an accurate measure to imply vector competence. The existence of factors such as midgut and salivary glands infection and scape barriers, host immunity,  etc., are components that ultimately interfere with the ability of a virus to be transmitted by the host and that are ignored when choosing to perform whole-body analyses rather than dissection and quantification of viral particles in a tissue-specific manner, with the inclusion of measuring viral particles in saliva expectorates as gold standard procedure in vector competence assays. Authors are strongly encouraged to remove this term from their title and throughout the manuscript, adapting accordingly. The conclusion provides a good example of how the language should be, rather than oversimplifying the concept of vector competence.

Response 1: Thank you for your comment. After consideration, we agree with the reviewer and  have replaced the term vector competence with vector infection rates” as well as, vector transmission with vector infection rates” throughout the entire manuscript.

The title of the manuscript was changed from “A BTV strain specific single gene that increases Culicoides vector competence” to “Identification of a BTV strain specific single gene that increases Culicoides vector infection rate”

Line (L): 106-107: could the authors please include a few referenced sentences on how common the issue of co-infection in ruminants is?

Response 2: Yes, a sentence was added (L109-111). “In areas, where several different BTV serotypes are simultaneously circulating, co-infection of ruminants with multiple serotypes has been reported previously.”

L 127-129/L172-177: These are good examples of how much attention the authors paid to their text, to make it accessible.

Response 3: Thank you for your kind comment.

L.155: could the authors please confirm the link? I got an error message saying the URL cannot be found, although the reoviridae.org domain still works.

Response 4: Thank you for reporting this issue, we double checked this link and it worked fine. However, we added “https://” to the link https://www.reoviridae.org/dsRNA_virus_proteins/ReoID/BTV-isolates.htm  for clarity.

The reoviridae.org still works, but the long-term plan is to rebuild this website. Some information regards our BTV isolates can be also found on the EVAg portal https://www.european-virus-archive.com/ .

  1. 284: Could the authors please elaborate on what is the commonly reported EIP for wtBTV under this temperature condition, to understand if 8dpi is a good collection point or not.

Response 5:  Previous studies have demonstrated that after infection of C. sonorensis with BTV at 25°C (day 0), viral load remains at or below the detection limit (day 2-4) and then rapidly increases (day 4-6) in the proliferation phase. The proliferation phase is temperature-dependent: it is shorter in higher temperature (2-3 dpi in 30 °C) and longer in lower temperature (~12 days at 20°C). Numerous studies investigating the EIP of BTV and other orbiviruses in Culicoides at different temperatures ( e.g.  Wittmann et.al 2002; Carpenter et.al 2011) highlighted that 8-days incubation at 25°C is a good incubation time point across numerous orbivirus strains to ensure viral replication, full virus dissemination and insect survival. Most importantly an eight-day incubation was also utilised successfully in oral infection experiments at the Pirbright Institute using the BTV wild type strains (Sanders et.al. 2021 plus unpublished data) that provided the foundation for the reverse engineered viral strains used here and therefore the incubation time of blood-fed Culicoides was kept consistent for better comparison.  

Sanders et. al 2021 is out on BioRxive https://www.biorxiv.org/content/10.1101/2021.08.09.455771v1.full

Wittann et al 2002 Effect of temperature on the transmission of orbiviruses by the biting midge, Culicoides sonorensis Medical and Veterinary Entomology

Carpenter et al 2011 Temperature Dependence of the Extrinsic Incubation Period of Orbiviruses in Culicoides Biting Midges Plos One

L 316: “One well as used…” typo?

Response 6: Change was made: “One well was used…” (L321)

L436 – Table 1: Could the information for VP2 be also included in this table?

Response 7: Table 1 represents only aa differences between reverse engineered rBTV-1 and rBTV-4 strains for high aa similarity segments; therefore, segment- 2 and segment-6 were excluded from this table. There are too many aa changes in seg-2 and seg-6 between both serotypes to list them in a table format. The addition of multiple changes occurring in seg-2 and -6 in the table 1 may be confusing. The purpose of the table was to identify the importance of aa changes in segments different than 2 and 6.

L454 – Figure 1: Can the authors please elaborate on why wtBTV1 and 4 were not included in parallel, as a reference for comparing viral replication efficiency between rescued and reassortant strains? I assume the authors decided to use this approach based on their %aa similarity results between rBTVs and wtBTVs, but given the importance of VP2 is to viral entry and attachment, and its low aa% similarity reported, wouldn't it be ideal to have had this included in the in vivo study as it was for the cell kinetics assay? This comment is also based on the results reported on L492-507 in which differences were observed for BTV-1. On a minor note, I would suggest moving away from the yellow color, as it makes it really hard to visualize the individual data points. I use and recommend a website called www.coolors.co as a really nice alternative to select for visually appealing color combinations/data viz.

Response 8: We compared full genome sequencing between rBTV and wtBTV isolates and found out that VP2 segment of rBTV-1 and wtBTV-1 was identical on the amino acid level as well as VP2 segment of rBTV-4 and wtBTV-4.  Any minor aa differences between wtBTV and rBTV were described in section 3.3 of the manuscript. We considered the sequencing analysis as sufficient to confirm that rBTV resemble wtBTV. However, wtBTV-1 and wtBTV-4 infection rate of Culicoides vector have been compared in our recent manuscript Sanders et al 2021 [submitted] and are in an agreement with the results from this study (high infection rate for BTV-4, low infection rate for BTV-1). Unfortunately, we are not in the position to perform any additional blood feeding experiments.

Yellow colour has been changed in Figure 1 as requested

L.476 – Figure 2:

Pt.1: Would be interesting if the authors had included 8dpi in their cell kinetics study to match the timepoint in which midges were collected to assess viral load. Any reasons why this was not performed?

Response 9: We decided to extract the estimated titres at 8 dpi from the curves for each strain (see table below) as requested.

Estimated log10 copies for eleven BTV strains in KC cells at eight days post infection.

strain

posterior median

95% credible limits

lower

upper

rBTV-1

6.7

6.1

7.4

rBTV-4

6.3

5.6

6.9

BTV-41S2S6S7

6.3

5.7

7.0

BTV-14S2

6.7

6.0

7.3

BTV-14S2S6S7

6.5

5.8

7.1

BTV-14S3

6.9

6.3

7.6

BTV-4­1S3

6.8

6.1

7.5

BTV-14S9

6.9

6.3

7.6

BTV-41S9

6.5

5.8

7.2

wtBTV-1 (MOR2006-06)

6.7

6.0

7.4

wtBTV-4 (MOR2004-02)

6.7

6.1

7.4

The upper asymptotes in the replication curves don't differ amongst the strains, but all strains haven't reached the asymptote by day 8. Estimated log copies are all generally higher than the levels in the insects. After consideration we don’t see the need to include this information in our manuscript.

Pt. 2: Can the authors please re-arrange this to first display the strains used to infect the midges, next to each other, and separated from the remainder, as to facilitate visual comparisons?

Response 10: Figures 2 & 3 were re-arranged as requested.

  1. 481 – Figure 3: I understand that the results show a maximum replication rate <10dpi, but for consistency, could the X-axis be up to 10 dpi to reflect all collection points?

Response 11: Figures 3 was changed as requested.

Table S4: Could the authors provide the results using the scientific notation (x power of 10) to facilitate the visualization?

Response 12: The scientific annotation was used in Table S4 and Table S5 as suggested.

Reviewer 2 Report

The paper by Ropiak et al. deals with the role of specific segments of BTV genome in influencing its ability to infect Culicoides biting midges of the species C. sonorensis belonging to a lab colony. Via a reverse genetic system, Authors tried to define the impact of different segments of BTV genome. The paper is original, interesting, well written, adequate in methods and statistics and in the expression of the obtained results.

Nevertheless, a very relevant general objection has to be made, regarding the core itself of the work. As a matter of fact, this work doesn’t deal with vector competence!!!! Hence, the whole manuscript should be modified, starting from the title, correctly declaring that the work regards infection rates, that is a quite different thing respect to competence. This is a very relevant problem and the paper, otherwise interesting and deserving publication, should not be published if Authors do not modify properly the whole text correctly, stating they are working on infection rates. The funny thing is that Authors themselves in introduction correctly explain what competence is (Lines 53-54) and that they are working on infection rates (Line 131), but along the whole manuscript, and in the title itself, they improperly use the term competence. They in no way evaluated the ability of the colony C. sonorensis to transmit the different natural and lab strains of the virus, hence they didn’t evaluate competence. Furthermore, at lines 65-69 the term competence is used in an even more questionable sense, increasing the confusion of the reader.     

I here down itemise come other changes that should be done and some points that should be addressed to make the manuscript deserving publication on Viruses:

  1. Line 3: “competence” should be changed into “infection rates”;
  1. Lines 51: Diptera and Ceratopogonidae not in italics. Only genus and species names have to be written in italics;
  2.  Line 57-62: this section refers to a general definition, not only regarding viruses transmitted by Culicoides. Hence it should be modified, otherwise the reader could assume that the reported explanations only refer to Culicoides and viruses they transmit;
  3. Lines 65-69: this ambiguous use of the term "competence" instead of easing comparisons is a possible cause of confusion. Authors should use another term to indicate the ability of BTV strains to replicate in midges. Moreover, see general comments regarding the improper use of the term competence along the whole manuscript;
  4. Lines 71: family name Reoviridae not in italics;
  5. Line 131: for the first time in the manuscript Authors correctly declare they are determining infection rates, rather than competence;
  6.  Line 333-334: not competence, infection rates;  
  7. Lines 603-605: this sentence is wrong. Something is lacking.

In conclusion, the paper by Ropiak et al. is interesting and well done and could be published, but only after addressing the general, relevant concern regarding the improper use of the term competence along the whole manuscript.   

Author Response

Response to Comments of Reviewers

Manuscript ID: viruses-1356271
Type of manuscript: Article
Title:
Identification of a BTV strain specific single gene that increases Culicoides vector infection rate

We would also like thank to the reviewers for their valuable comments, which have been largely implemented and improved this manuscript. Please find our replies to the specific comments (in italics).

Reviewer 2:

The paper by Ropiak et al. deals with the role of specific segments of BTV genome in influencing its ability to infect Culicoides biting midges of the species C. sonorensis belonging to a lab colony. Via a reverse genetic system, Authors tried to define the impact of different segments of BTV genome. The paper is original, interesting, well written, adequate in methods and statistics and in the expression of the obtained results.

Nevertheless, a very relevant general objection has to be made, regarding the core itself of the work. As a matter of fact, this work doesn’t deal with vector competence!!!! Hence, the whole manuscript should be modified, starting from the title, correctly declaring that the work regards infection rates, that is a quite different thing respect to competence. This is a very relevant problem and the paper, otherwise interesting and deserving publication, should not be published if Authors do not modify properly the whole text correctly, stating they are working on infection rates. The funny thing is that Authors themselves in introduction correctly explain what competence is (Lines 53-54) and that they are working on infection rates (Line 131), but along the whole manuscript, and in the title itself, they improperly use the term competence. They in no way evaluated the ability of the colony C. sonorensis to transmit the different natural and lab strains of the virus, hence they didn’t evaluate competence. Furthermore, at lines 65-69 the term competence is used in an even more questionable sense, increasing the confusion of the reader.     

Response 1: Thank you for your comment. After consideration, we agree with the reviewer and have replaced the term vector competence with vector infection rate(s)” as well as, vector transmission with vector infection rate(s)” throughout the entire manuscript.

The title of the manuscript was changed from “A BTV strain specific single gene that increases Culicoides vector competence” to Identification of a BTV strain specific single gene that increases Culicoides vector infection rate”

Line 65-69 was deleted.

I here down itemise come other changes that should be done and some points that should be addressed to make the manuscript deserving publication on Viruses:

  1. Line 3: “competence” should be changed into “infection rates”;

Response 2: The title of the manuscript has been changed and word “competence” was replaced with “infection rate” (Line 3).

  1. Lines 51: Diptera and Ceratopogonidae not in italics. Only genus and species names have to be written in italics;

Response 3:  The italics have been removed (Line 51).

  1.  Line 57-62: this section refers to a general definition, not only regarding viruses transmitted by Culicoides. Hence it should be modified, otherwise the reader could assume that the reported explanations only refer to Culicoides and viruses they transmit;

Response 4: Thank you, we incorporated this suggestion.

We replaced:

"Vector competence is commonly expressed as the percentage of individual Culicoides midge that develop a fully transmissible infection following the uptake of a virus-containing blood meal. Culicoides spp. vector competence depends on a range of factors, including the genetic susceptibility of the midge species…."

With:

"Vector competence is commonly expressed as the percentage of individual arthropod that develops a fully transmissible infection following the uptake of a virus-containing blood meal. In the case of Culicoides spp. vector competence depends on a range of factors, including the genetic susceptibility of the midge species…." (Lines 57-69).

  1. Lines 65-69: this ambiguous use of the term "competence" instead of easing comparisons is a possible cause of confusion. Authors should use another term to indicate the ability of BTV strains to replicate in midges. Moreover, see general comments regarding the improper use of the term competence along the whole manuscript;

Response 5: Line 65-69 was deleted.  “High vector competence term was replaced with “high vector infection rate(s)as well as, vector transmission with vector infection rate(s)throughout the entire manuscript.

  1. Lines 71: family name Reoviridae not in italics;

Response 6: According to International Committee on Taxonomy of viruses (https://talk.ictvonline.org/ictv-reports/ictv_online_report), the virus family is always written in italics.

  1. Line 131: for the first time in the manuscript Authors correctly declare they are determining infection rates, rather than competence;

Response 7: Thank you for your comment.

  1.  Line 333-334: not competence, infection rates;  

Response 8: Thank you, we incorporated this suggestion here (Lines 340-341) and through the text ( Line 3, 298, 301, 436, 448,450, 451, 452, 456, 458, 460, 463, 523, 528, 540, 574, 581, 583, 587, 603, 629)

  1. Lines 603-605: this sentence is wrong. Something is lacking.

Response 9: We replaced:

"The substitutions in Seg-6 and Seg-7 of BTV-14S2S6S7 were the same as in wtBTV-4 (cell passage E1/BHK4) possibly indicating a BTV adaptation to a mammalian cell line".

with:

"The substitutions of aa in Seg-6 and Seg-7 of BTV-14S2S6S7 were the same as in wtBTV-4 (cell passage E1/BHK4) possibly indicating a BTV adaptation to a mammalian cell line". (Lines 604-607)

In conclusion, the paper by Ropiak et al. is interesting and well done and could be published, but only after addressing the general, relevant concern regarding the improper use of the term competence along the whole manuscript.   

Response 10: thank you for your comment, the nomenclature has been changed thought the manuscript.

Round 2

Reviewer 2 Report

The manuscript has been properly modified after the first review round, and it now deserves publication.

The only mistake, at line 73, as already highlighted during the first review, Reoviridae not in italics!!!